# TikTok and the Art of Personalization: Investigating Exploration and Exploitation on Social Media Feeds

## ABSTRACT

Recommendation algorithms for social media feeds often function as black boxes from the perspective of users. We aim to detect whether social media feed recommendations are personalized to users, and to characterize the factors contributing to personalization in these feeds. We introduce a general framework to examine a set of social media feed recommendations for a user as a timeline. We label items in the timeline as the result of exploration vs. exploitation of the user's interests on the part of the recommendation algorithm and introduce a set of metrics to capture the extent of personalization across user timelines. We apply our framework to a real TikTok dataset and validate our results using a baseline generated from automated TikTok bots, as well as a randomized baseline. We also investigate the extent to which factors such as video viewing duration, liking, and following drive the personalization of content on TikTok. Our results demonstrate that our framework produces intuitive and explainable results, and can be used to audit and understand personalization in social media feeds.

ACM Reference Format:
Anonymous Author(s). 2023. TikTok and the Art of Personalization: Investigating Exploration and Exploitation on Social Media Feeds. In *Proceedings of Make sure to enter the correct conference title from your rights confirmation emai (WebConf '24).* ACM, New York, NY, USA, 9 pages. https://doi.org/XXXXXXX.XXXXXXX

## 1 INTRODUCTION

Social media platforms traditionally provided their end-users with content that was ordered chronologically and came from users' connections. However, in recent years, these platforms have started employing recommendation systems to select the content shown on their audience feeds. Moreover, these algorithmic feeds are personalized for the end-users to ensure that the users will be served with the content they are most likely interested in.

At the same time, with the rapid rise of the TikTok platform, we are witnessing an extremely popular trend centered around short format videos (30-60 seconds) on social media platforms. The combination of these short videos, paired with an algorithmically driven feed, can potentially cause adverse effects on users since the recommendation algorithm makes several short format content recommendations over a short period. In recent years, researchers have raised concerns that excessive personalization of algorithmic

social media feeds can potentially trap people in filter bubbles and echo chambers [21]. This can lead to a variety of harms ranging from driving young users to depression and self-harm to creating highly polarized, radicalized, and ideologically fragmented societies [7, 14, 22].

In response, policymakers stepped in to address these growing concerns. In fact, the recently passed EU legislation, the Digital Services Act (DSA) [5] emphasizes the importance of algorithmic transparency and calls for audits of algorithmic feeds. Hence, there is a pressing need to investigate these feeds to understand the extent of personalization and the possible effects this personalization can have on users.

To bridge this gap, with an emphasis on short-format videos, our work focuses on analyzing and auditing TikTok's personalized social media feed that combines short-format content and algorithmic recommendations. We focus on the following research questions:

- **RQ1:** Given a sequence of content recommendations from a user's feed, how can we detect which recommendations are the result of personalization?
- **RQ2:** How do certain factors influence the extent of user personalization on TikTok?

We propose and validate a framework that allows us to model and assess the extent of user personalization on users' social media feeds. Particularly, given a set of social media feed attributes that include content, user, and engagement attributes, we design and implement a framework to assess which video recommendations are the result of personalization (i.e., *exploit recommendations*) and which are not (i.e., *explore recommendations*). We validate and demonstrate the applicability of our framework on a dataset of real traces from TikTok users collected by [24], as well as other baselines, including traces obtained from automated accounts on TikTok and a randomized baseline.

**Contributions.** The contributions of this work are two-fold. First, the proposed framework is flexible and can be used to audit any personalized social media feed, including those of other short format video sites like Instagram and YouTube Shorts and those of traditional content feed sites like Facebook and Twitter. Also, we believe our framework can be used as part of algorithmic transparency and auditing systems that aim to provide feedback to users on how personalized their feeds are and the underlying reasons for getting recommended specific content.

Second, our methods applied to TikTok's For You feed shed light on its recommendation algorithm. We find, for example, that the algorithm exploits users' interests in between 30% and 50% of all recommended videos in the first thousand videos of users' tenure on TikTok. We introduce the notion of a *personalization score* and observe that this score can indeed estimate the extent of personalization in users' feeds. We also examine personalization factors on TikTok, and our results show that liking and following are the primary drivers of content personalization for users.

**Broader Perspective and Ethical Considerations.** We obtained approval from our institution's Ethical Board Review Committee before conducting any data collection/analysis. We emphasize that the real-world traces dataset, obtained by previous work by Zannettou et al. [24], was collected after obtaining explicit consent from the participants before data donation. Additionally, the video metadata collection focuses on publicly accessible videos at the time of data collection (i.e., we do not have any data about deleted videos or videos from uploaders with private accounts). When conducting our data analysis, we follow standard ethical guidelines [15] like not attempting to de-anonymize participants and performing our analysis on aggregate. We believe our work ultimately benefits end-users of social media platforms by supporting transparency and algorithmic auditing.

## 2 BACKGROUND & RELATED WORK

TikTok is a social media platform with increasing popularity; in 2022, TikTok was the most downloaded mobile application worldwide with 672M downloads [19]. TikTok heavily relies on two features that make the platform stand out from other social media platforms; short-format videos and an algorithmic recommender system that offers an endless stream of video recommendations to users. TikTok offers many traditional social networking features such as: 1) users can follow other users, thus forming a social network, and 2) users can engage with a video by liking it, commenting on it, sharing it, or marking it as a favorite video. Recently, reports have emerged (e.g., [9, 17]) that indicate TikTok's recommendation algorithm is very effective in inferring users' interests and making recommendations that the users eventually like or engage with.

**Previous Work.** A substantial body of work leverages automated accounts and qualitative analyses to understand and analyze TikTok's algorithmic recommendations. Boeker and Urman [2] explore the effect of various features on the TikTok algorithm personalization. Using automated accounts (i.e., bots), they investigate how features such as the like/follow features, region, language, and how much time users spent on specific content, affect the extent of personalization. Overall, they find that the follow feature exerts the most influence on the extent of personalization, followed by the like feature. Journalistic investigative efforts from the Wall Street Journal [18] created over 100 automated accounts to understand what features (i.e., follow, like, share, watch time) affect algorithmic recommendations on TikTok. Their investigation finds that one of the signals alone, the watch time (i.e., time a user spends on a video), provides a strong signal to the algorithm and substantially affects algorithmic recommendations. They find that for automated accounts that expressed an interest in problematic content such as depression (as determined by the watch time the user spent on video with such content) often lead users down an algorithmic rabbit hole of problematic content. Klug et al. [6] undertake a mixed-methods analysis of TikTok's algorithm and find that videos with high engagement are more likely to be recommended by the algorithm. At the same time, they find that using very popular hashtags (e.g., #foryou) does not increase the likelihood of a video being recommended by the algorithm. Bandy and Diakopoulos [1] explore the role of TikTok's recommendation algorithm in promoting call-for-action videos with a case study on the Tulsa rally. Their analysis shows

that the amplification of call-for-action videos is not systematic and is rather likely due to the videos having an increased engagement. Lee et al. [8] undertake a qualitative analysis of 24 interviews with TikTok users, aiming to explore how algorithmic personalization affects users' perceptions. Their work highlights that TikTok users can identify parts of their identity via the algorithmic recommendation of the "For You" page and that their behavior can shape the algorithm's ability to reflect their diverse interests. Simpson and Semaan [16] perform an interview study by recruiting 16 LGBTQ+ TikTok users, aiming to understand these users' interactions and encounters on TikTok; they find that TikTok's algorithm creates contradictory identity spaces that reaffirm LGBTQ+ identities while simultaneously violating intersections of user identities.

**Remarks.** Previous work attempted to demystify algorithmic recommendations by heavily relying on traces obtained exclusively from automated accounts or through the lens of users' responses. While these previous efforts yield important insights, we argue that traces from automated accounts may lack the authenticity and diversity of traces from real users. Additionally, these efforts are limited as user self-reports might introduce biases or discrepancies in the results [3, 12, 23]. In our work, we overcome these challenges and analyze algorithmic recommendations through the lens of traces from real TikTok users. At the same time, we complement our analysis with traces from automated accounts that act as baselines for our analyses. To the best of our knowledge, our framework is the first that focuses on understanding the interplay between exploration and exploitation in TikTok recommendations. Additionally, we propose a general framework that can be applied to other algorithmic-powered social media feeds.

## 3 DATASETS

This section describes the two datasets used in our analysis. We use a dataset that includes traces from real TikTok users and a dataset that includes traces from automated bots.

### 3.1 Dataset from Real Users

Our dataset of traces from real TikTok users, is based on previous work by Zannettou et al. [24]. The authors relied on the EU's General Data Protection Regulation (GDPR), particularly the right of access by data subjects. They implemented a privacy-preserving data donation system and recruited 347 TikTok users that donated their TikTok traces. Each trace provides a comprehensive view of the user's actions on TikTok, including the user's viewing history, like history, search history, follow history, etc. (see [24] for a comprehensive list of all the fields included in the datasets). Additionally, for each video referenced in the traces from real-world TikTok users, the authors collected additional metadata, as the traces from TikTok include only video identifiers. To do this, they used an unofficial Python wrapper for the TikTok API [20], which allowed them to collect metadata for each video, including the video description, the video hashtags, statistics about the video, etc. Overall, the dataset includes 4.9M videos viewed 9.2M times by 347 recruited TikTok users. Note, that only 4.1M videos have associated video metadata (the rest of the videos were either deleted or the uploader made their account private, by the time of the data collection).

**Table 1: Overview of bot configurations for obtaining the bot traces. The table shows the probabilities of the bots watching a video till the end (Watch), skipping a video (Skip), liking a video (Like), and following the video creator (Follow).**

|  | $\mathcal{P}(Watch)$ | $\mathcal{P}(Skip)$ | $\mathcal{P}(Like)$ | $\mathcal{P}(Follow)$ |
|---|---|---|---|---|
| Bot 1 | 0 | 1 | 0 | 0 |
| Bot 2 | 1 | 0 | 0 | 0 |
| Bot 3 | 0.5 | 0.5 | 0 | 0 |
| Bot 4 | 1 | 0 | 0.5 | 0.5 |
| Bot 5 | 0.5 | 0.5 | 0.5 | 0.5 |

## 3.2 Dataset from Automated Bots

We complement our real-world user traces dataset with a bot dataset, which we refer to as `simulated-bot`, that comprises Tiktok's `For You` recommendations for a set of automated accounts on TikTok. Our dataset consists of traces generated by five automated bots; for each bot, we create a new TikTok account, and we use a random date of birth and a unique email address. No other personal information, such as gender or location is provided when creating the accounts. Each account is controlled by an automated bot with a pre-defined policy that dictates the bot's behavior. Table 1 reports the policies of our bots. We implement the bots and their policies using the PlayWright framework [13]. Each bot visits Tiktok's `For You` feed via TikTok's Web interface and watches 1,000 videos according to the bot's pre-defined policy. Also, we perform a video metadata collection for each video that a bot encounters using an unofficial Python wrapper for the TikTok API [20]. The video metadata collection is done in real-time, ensuring we obtain video metadata for all videos in our dataset. We run our bot dataset collection between December 28, 2022, and January 17, 2023.

## 4 RQ1: DETECTING USER PERSONALIZATION

This section presents our modeling framework for detecting videos resulting from user personalization.

## 4.1 Social Media Feed Attributes

We begin by describing three broad categories of data attributes that we use: *content*, *user*, and *engagement*. These attributes are readily extracted from most social media feed data and form the basis of our framework to investigate the extent of user personalization in a social media feed. Note that the attributes we describe are not an exhaustive list of all the possible attributes but an intuitive and comprehensive set we curate that applies to most social media feeds.

**Content Attributes.** Given a set of $N$ chronologically ordered recommendations $R = \{r_1, r_2, r_3, \ldots, r_N\}$ each recommendation item $r_i$ has a set of attributes that provide details about its content.

- *Recommendation Index:* For any set of $N$ chronologically ordered recommendations $R = \{r_1, r_2, r_3, \ldots, r_N\}$, the recommendation index, $i \in \mathbb{Z}^+$ (i.e., $1 \leq i \leq N$) is the index of each recommendation item in sequential order.
- A set of $M$ hashtags included in the recommendation's description $H = \{h_1, h_2, h_3, \ldots, h_M\}$.

- A user, $u_C$ that created the content that is recommended. For example, on user-generated video platforms, this corresponds to the video uploader.

**User Attributes.** Each user, $u$ has a set of attributes that pertain to macroscopic user behaviors such as the accounts that are followed and the user's topics of interest.

- A set of $K$ other users, $U_F = \{u_1, u_2, u_3, \ldots, u_K\}$ that are followed.
- A set of $X$ interests, $I_P = \{h_1, h_2, h_3, \ldots, h_X\}$ as determined by the most popular hashtags in all recommendations to user $u$.
- A set of $Y$ interests, $I_D = \{h_1, h_2, h_3, \ldots, h_Y\}$ as determined by hashtags that user $u$ specifies directly with the platform.

**Engagement Attributes.** For every recommendation item - user pair $(r_i, u)$, we have a set of attributes that provide information about user $u$'s engagement with the recommendation item, $r_i$. We use the subscript $r_i, u$ to denote the engagement attributes of the recommendation item - user pair.

- *Timestamp:* The timestamp, $\theta_{r_i,u}$ when the recommended item $r_i$ was viewed by the user $u$.
- *Engagement Duration:* The duration (in seconds), $t_{r_i,u}$ the user $u$ engaged with (watched for videos) the recommended item $r_i$.
- *Liking:* We use a boolean, $\text{liked}_{r_i,u}$ to denote whether the user $u$ liked the recommended item $r_i$, and if $\text{liked}_{r_i,u} = \text{True}$, we record the timestamp the item was liked.
- *Following:* We use a boolean, $\text{followed}_{r_i,u}$ to denote whether the content creator of recommended item $r_i$ is followed by the user $u$, and if $\text{followed}_{r_i,u} = \text{True}$, we record the timestamp the creator was followed.
- *Sharing:* We use a boolean, $\text{shared}_{r_i,u}$ to denote whether the item $r_i$ was shared by the user $u$, and if $\text{shared}_{r_i,u} = \text{True}$, we record the timestamp $r_i$ was shared.
- *Favoriting:* We use a boolean, $\text{favorited}_{r_i,u}$ to denote whether the item $r_i$ was favorited by the user $u$, and if $\text{favorited}_{r_i,u} = \text{True}$, we record the timestamp $r_i$ was favorited.

## 4.2 Exploitation vs. Exploration Framework

Given a user $u$ and a set of $N$ chronologically ordered recommendations $R_u = \{r_1, r_2, r_3, \ldots, r_N\}$, we aim to demystify which recommendations are the result of user personalization and which are not. We define an *exploitation* recommendation as a recommendation that is personalized based on the user's inferred interests or previous actions. On the other hand, we define an *exploration* recommendation as a recommendation that is *not* the result of user personalization, but due to the algorithm trying to explore if the user might like a specific – and often new or different – topic.

We assume that certain items recommended to a user are related to prior user actions or items recommended to that user. We model a user $u$'s trace of viewing history as a timeline where each item $r_i$, has an associated timestamp $\theta_{r_i,u}$ for user $u$. We evaluate the extent of personalization of each item-user pair $(r_i, u)$ i.e., the degree of personalization of each recommended item at a specific point in time. To do this, we leverage content, user, and engagement attributes to determine whether (and the extent to which) a recommended item $r_i$ for user $u$ is related to previously recommended items $r_j$ in $u$'s feed, where $\theta_{r_j,u} < \theta_{r_i,u}$.

### 4.2.1 Global and Local Features.
We define a set of features, $F = \{f_1, f_2, \ldots, f_n\}$, derived from the attributes available in a user's

recommended item timeline. Our framework is flexible and readily generalizable, as it allows the addition or removal of features based on the use case.

We use these features, $F$ to evaluate the connectedness of items in each user's timeline by specifying an *activation* condition for each feature; i.e., if the condition specified by the feature is satisfied for an item, then we mark the corresponding item as *activated*. Our framework defines features such that the activation condition labels an item as an *exploit* recommendation.

**Local features** model relationships between items within a specific *temporal window* of size $W$. For example, for item-user pair $(r_i, u)$, the local feature likes_hashtag_local considers preceding items $r_j$ in the temporal window $W$, $r_j \in \{r_{i-1}, \ldots, r_{i-W}\}$, and *activates* the item $r_i$ if $\text{liked}_{r_j,u} = \text{True}$ and $r_j$ has a hashtag in common with $r_i$. In other words, in this example, we label a recommendation item, $r_i$ as *exploit* if the user liked a previous item (within $W$) that shares a hashtag with $r_i$.

**Global features** are general attributes that capture personalization at a macroscopic scale for a particular user. For example, for item-user pair $(r_i, u)$, the global feature following_global considers all previously recommended items $r_j$, such that $\theta_{r_j,u} < \theta_{r_i,u}$ in $u$'s feed that have $\text{following}_{r_j,u} = \text{True}$, and *activates* the item $r_i$ if $r_j$ has a hashtag in common with $r_i$. In other words, in this example, we label a recommendation item, $r_i$ as *exploit* if a user followed the recommended item's creator anytime *before* they were recommended $r_i$.

*4.2.2 Labeling Recommendations.* For each recommended item-user pair in a user $u$'s timeline we consider a set of $d$ features $F = \{f_1, f_2, \ldots, f_d\}$.

- We label item-user pair $(r_i, u)$ as an *Exploit* recommendation if any of the local or global features, $F = \{f_1, f_2, \ldots, f_d\}$ for that recommendation item satisfy the activation condition for $(r_i, u)$.
- If *none* of the features $F = \{f_1, f_2, \ldots, f_d\}$ used result in $(r_i, u)$ being activated, then that item-user pair is marked as an *Explore* recommendation.

*4.2.3 Personalization Metrics.* We define three metrics in our analysis which serve as quantitative measures to study the extent of user personalization in a social media feed. We assume the recommendation labeling method outlined above using a set of features $F = \{f_1, f_2, \ldots, f_d\}$ to label each item-user pair, $(r_i, u)$ in user $u$'s timeline as exploit or explore.

**User exploit fraction.** Given a set of recommendations for a certain user, $u$ a recommendation index, $i$ and a window $W$, we define the user's *exploit* fraction at recommendation index $i$ to be the fraction of items in the window $W$ that are marked as *exploit* recommendations. Consider items $r_j \in \{r_{i-1}, \ldots, r_{i-W}\}$ recommended to user $u$, and let $N_{ui}$ correspond to the number of $r_j$'s that are labelled *exploit*. Then for the recommendation index, $i$ we denote the user exploit fraction as $\alpha_{ui} = \frac{N_{ui}}{W}$

**Mean exploit fraction.** Given a set of $m$ users, their corresponding recommendation feeds, and a recommendation index, $i$, another quantity of interest is the mean user exploit fraction, which we define as the arithmetic mean of the exploit fractions of all $m$ users in the set, at recommendation index $i$. We denote the mean user exploit fraction by $\overline{\alpha}_i = \frac{1}{m} \sum_{u=1}^m \alpha_{ui}$

**Personalization score.** We introduce the concept of a personalization score, where the core idea is to ascertain "how personalized" a specific recommendation item $r_i$ is for a user $u$. Given an item-user pair, $(r_i, u)$, and $m$ total user timelines, we estimate the extent of personalization, $\rho(r_i, u)$. This is achieved by calculating the number of user timelines in which the recommended item $r_i$, when inserted at recommendation index $i$, would have the same label (Explore or Exploit) as it did for user $u$. Assuming that item $r_i$ is marked Exploit (or Explore) and has the same label Exploit (or Explore) in $k$ other user timelines when inserted at recommendation index $i$, we then define the personalization score for item-user pair $(r_i, u)$ as: $\rho(r_i, u) = 1 - \frac{k}{m}$. Note that the personalization score is not a symmetric measure, in general. Intuitively we expect items labeled exploit to have a higher personalization score, assuming they were tailored recommendations, since these items would have a lower probability of being marked as exploit in other user timelines. In contrast, we expect the personalization score of items labeled explore to be low, since these items are likely to also be marked as explore in other user timelines.

*4.2.4 Framework Specification.* Here, we describe the parameters used to specify our modeling framework.
1. A set of $m$ users for whom we have data about their timeline of recommended items with social media feed attributes specified.
2. The sample size, $N$. We include the first $N$ chronologically ordered recommendation items from a user's feed.
3. A temporal window of size $W$, that measures the local features by considering up to $W$ recommendation items in the past.
4. The interest radii, $X$ and $Y$, which define how many of the top interests (hashtags) of each user we consider.
5. A set of $d$ features, $F = \{f_1, f_2, \ldots, f_d\}$ that have been pre-selected to model user personalization in the feed.

## 4.3 Feature Selection and Framework Evaluation

In this section, we discuss our baselines, feature selection technique, and how we evaluate our framework.

*4.3.1 Baselines.* Intuitively, in a randomized set of $N$ recommendations $\{r_1, r_2, r_3, \ldots, r_N\}$, there is no user personalization since the recommended items are (ideally) unrelated to each other. However, our features could be activated due to inherent noise in the data - for example, several random videos might have certain common hashtags. Consequently, we use a randomized baseline to select the features that minimize the noise as measured by randomized data. Also, we use a second baseline to evaluate our framework's results and explanatory power. We describe the two baselines here.

**Randomization by recommendation index.** We use $m$ real user traces to create our first randomized baseline; for each recommendation index, we randomly permute all items at that index across all user timelines. We use these $m$ randomized user traces as a baseline for feature selection. In practice, we repeat the above randomization several times, and report results averaged across all the random samples. We refer to this randomized baseline dataset as index-randomized.

**Automated bot traces.** We generate automated bot timelines using the bots described in Section 3.2. We use these bot traces to

**Table 2: Sample social media feed features for item-user pair $(r_i, u)$, with creator $u_C$, and temporal window $W$.**

| Feature | Activation condition for item-user pair $(r_i, u)$ to be marked *exploit* |
|---|---|
| generic_hashtag_local | $r_i$ has a hashtag in common with any preceding item in the temporal window $W$, $r_j \in \{r_{i-1}, \ldots, r_{i-W}\}$. |
| generic_creator_local | $r_i$ has the same creator, $u_C$ as any preceding item in the temporal window $W$, $r_j \in \{r_{i-1}, \ldots, r_{i-W}\}$. |
| likes_hashtag_local | $r_i$ has a hashtag in common with any item in the temporal window $W$, $r_j \in \{r_{i-1}, \ldots, r_{i-W}\}$ and liked$_{r_j,u}$ = True. |
| likes_creator_local | $r_i$ has the same creator, $u_C$ as any item in the temporal window $W$, $r_j \in \{r_{i-1}, \ldots, r_{i-W}\}$ and liked$_{r_j,u}$ = True. |
| watched_hashtag_local | $r_i$ has a hashtag in common with any item in the temporal window $W$, $r_j \in \{r_{i-1}, \ldots, r_{i-W}\}$ and $t_{r_j,u} \geq 100\%$. |
| watched_creator_local | $r_i$ has the same creator, $u_C$ as any item in the temporal window $W$, $r_j \in \{r_{i-1}, \ldots, r_{i-W}\}$ and $t_{r_j,u} \geq 100\%$. |
| shares_hashtag_local | $r_i$ has a common hashtag with any item in the temporal window $W$, $r_j \in \{r_{i-1}, \ldots, r_{i-W}\}$ and shared$_{r_j,u}$ = True. |
| shares_creator_local | $r_i$ has the same creator, $u_C$ as any item in the temporal window $W$, $r_j \in \{r_{i-1}, \ldots, r_{i-W}\}$ and shared$_{r_j,u}$ = True. |
| favoriteVideos_hashtag_global | $r_i$ has at least one hashtag in common with *any* prior item $r_j$ in $u$'s feed that has favorited$_{r_j,u}$ = True. |
| favoriteVideos_creator_global | $r_i$ has a creator in common with *any* prior item $r_j$ in $u$'s feed that has favorited$_{r_j,u}$ = True. |
| following_global | $r_i$ has at least one hashtag in common with *any* prior item $r_j$ in $u$'s feed that has following$_{r_j,u}$ = True |
| inferred_interests_global | $r_i$ has at least one common hashtag with $I_P = \{h_1, \ldots, h_X\}$, the $X$ most popular hashtags in all recommendations to $u$. |

help validate the explanatory power of our results by comparing our results in terms of personalization metrics with those of the automated bots. Intuitively, we expect these traces to display a higher degree of personalization than the randomized baseline but a lower degree than real user data. We refer to the bot baseline dataset as simulated-bot.

*4.3.2 Feature Selection.* We assume we have a set of $n$ features $F' = \{f_1, f_2, \ldots, f_n\}$, and the goal is to choose the top-$d$ features $F = \{f_1, f_2, \ldots, f_d\}$ to label each item-user pair, $(r_i, u)$ in user $u$'s timeline as exploit or explore. We define the *signal-noise ratio* of a feature (or set of features) to be the ratio of the feature's mean exploit fraction in real user data to the mean exploit fraction in randomized user traces. We use the signal-noise ratio to measure feature importance in the following feature selection process.

(1) We examine the recommendation item and user data, and consider the different content, user, and engagement attributes to compile a comprehensive list of $n$ potential global and local features, $F' = \{f_1, f_2, \ldots, f_n\}$ for labeling recommendation items. For reference, we specify a sample set of features applicable to a (TikTok) video recommendation feed in Table 2.

(2) Consider each feature $f_a \in F'$ and label each item-user pair, $(r_i, u)$ as exploit or explore using only this feature $f_a$. We then observe the mean exploit fraction, $\overline{\alpha}_i$ for each recommendation index $i \in 1, \ldots, N$. We repeat the experiment for the $m$ randomized user timelines in index-randomized, since the $\overline{\alpha}_i$ in the randomized timelines captures the noise-floor of feature $f_a$.

(3) Rank the features in descending order of their signal-noise ratio and note their mean exploit fraction in index-randomized. From these feature rankings, we choose the top $d' \leq n$ features with the highest signal-noise ratio that are below a suitable noise threshold, $\tau$ in index-randomized. The noise threshold, $\tau$ can be chosen by inspecting the distribution of $\overline{\alpha}_i$ in index-randomized across all features.

(4) For each of the $d'$ subsets of $d' - 1$ features we label each item-user pair, $(r_i, u)$ as exploit or explore if any of the features in the subset considered are satisfied. Again, we observe the mean exploit fraction, $\overline{\alpha}_i$ for each recommendation index $i \in 1, \ldots, N$ and repeat the same for index-randomized. We then rank the subsets by signal-noise ratio and recursively remove features

until we have a set of top-$d$ features $F = \{f_1, f_2, \ldots, f_d\}$. Note that we don't specify $d$ in our procedure, but choose a $d$ that gives the best trade-off between model complexity and signal-noise ratio.

*4.3.3 Evaluating Results.* We leverage the bot dataset, simulated-bot generated from automated bots to evaluate our framework and the efficacy of the model. For each recommendation index, we compare the mean exploit fraction from our framework with the mean exploit fraction in simulated-bot. Since the automated bots make randomized choices with independent probabilities, intuitively we expect our metrics to reveal a lower level of personalization associated with the bot-simulated data.

## 4.4 Experimental Setup for TikTok

Here, we describe how we configure our framework to assess user personalization on TikTok video recommendation social feeds. Due to space constraints we describe how we preprocessed the TikTok dataset in Appendix A.

*4.4.1 Instantiating the Framework for TikTok.*
- From our TikTok dataset of 347 users, we filter out users with fewer than 1000 recommendations, and consequently consider video feed recommendations for $m = 220$ users, such that we have $N = 1000$ chronologically ordered video recommendations $R_u = \{r_1, r_2, \ldots, r_{1000}\}$ for each user $u$.
- We use a temporal window of size $W = 50$, and an interest radius $X = Y = 25$ such that we consider the top-25 hashtags for each user in both the popular hashtag set $I_P = \{h_1, h_2, h_3, \ldots, h_{25}\}$ and the specified hashtag set $I_D = \{h_1, h_2, h_3, \ldots, h_{25}\}$. We performed a sensitivity analysis to tune these parameter values for our TikTok dataset.
- Using the feature selection technique outlined previously, we select the following set of $d = 7$ features that have the best signal-noise ratio, $F = \{$generic_creator_local, likes_hashtag_local, likes_creator_local, watched_hashtag_local, watched_creator_local, favoriteVideos_hashtag_global, following_global$\}$.
- We create $m = 220$ timelines, with items in each user's timeline corresponding to the video-user pairs, $(r_i, u)$ for that user. Then

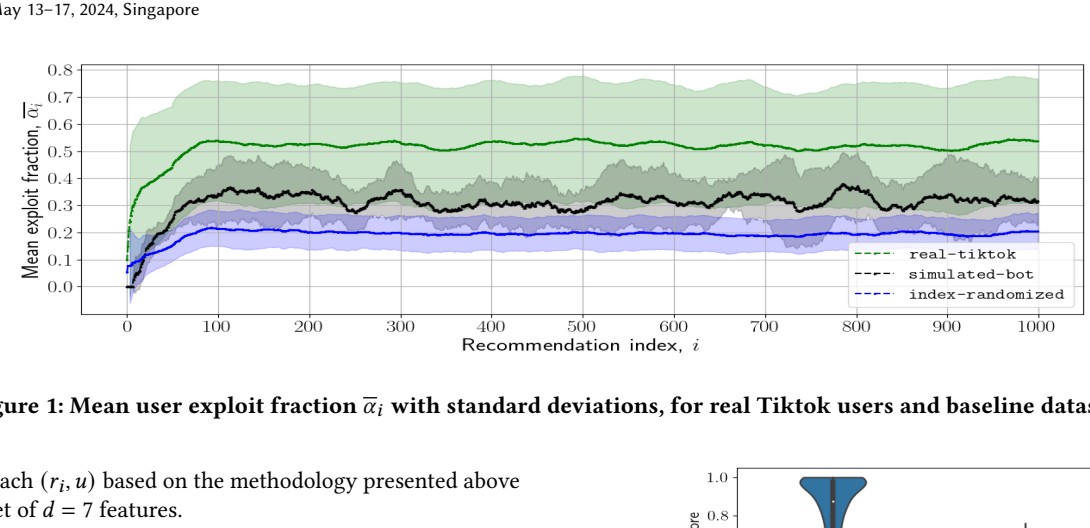

**Figure 1: Mean user exploit fraction $\overline{\alpha}_i$ with standard deviations, for real Tiktok users and baseline datasets.**

we label each $(r_i, u)$ based on the methodology presented above and the set of $d = 7$ features.

## 4.5 Results and Discussion

We use our framework and label all videos in the $m = 220$ user time-lines as exploit or explore recommendations, and we compute each user's exploit fraction for each recommendation index between $i \in 1, \ldots, 1000$. Then, we aggregate our analysis across the $m = 220$ individual users and obtain the mean user exploit fraction, averaged across all users in the TikTok dataset, for each recommendation index. We repeat the same procedure for the randomized baseline, `index-randomized` and the bot traces, `simulated-bot`. We visualize the mean user exploit fraction for the datasets in Figure 1.

First, we observe that the mean exploit fraction, $\overline{\alpha}_i$ for both `real-tiktok` and both baselines, initially increases steadily for the first few videos. We attribute the steady increase to (a) the temporal window $W$, since the first few videos recommended to a user do not have a full window of past videos and hence exhibit a lower exploit fraction; and (b) potentially to TikTok's algorithm ability to infer user interests and behavior, hence exploiting the users' interests to a greater extent.

The mean user exploit fraction is relatively stable for recommendation indices with $i > 100$. For `index-randomized`, this implies that the level of noise captured remains constant over time which is an expected result in a randomized timeline. This is also the expected result for `simulated-bot` since the automated bot traces represent timelines derived from random user behaviors. For the real user timelines, the stability of $\overline{\alpha}_i$ indicates that the TikTok recommendation algorithm recommends videos to users with a relatively constant level of personalization.

*4.5.1 Comparison with Baselines.* We validate our results using data from the automated bot traces, `simulated-bot`. We observe that the mean exploit fraction, $\overline{\alpha}_i \geq 50\%$, for `real-tiktok` dataset across most of the 1000 recommendation indices and differs significantly from that of both baseline traces. In contrast, we observe that the mean exploit fraction of the baselines is on average 31% for `simulated-bot` and 20% for `index-randomized`. Since the bots perform user behaviors randomly, intuitively we expect these time-lines to have a lower degree of personalization than real users. This is evident from Figure 1, where we observe that the automated bot traces have only about 60% of exploit videos compared to real user traces. Under the assumptions of our framework, we observe

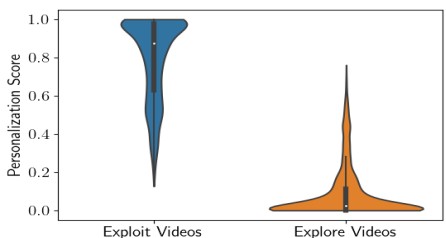

**Figure 2: Distribution of Personalization scores for Explore and Exploit labeled videos.**

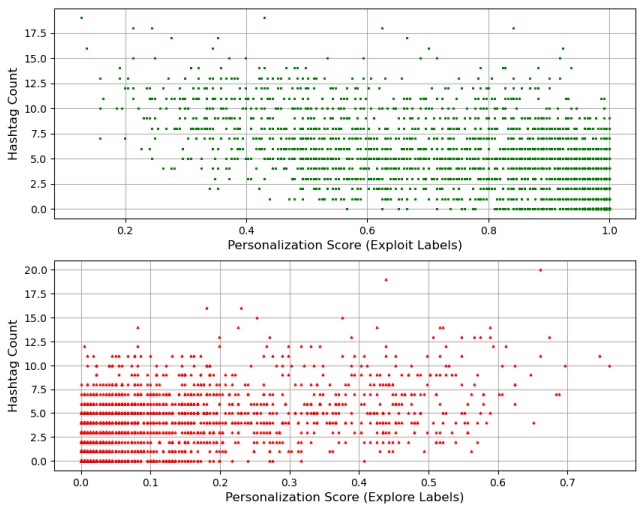

**Figure 3: Scatterplot of Personalization scores vs. Hashtag counts for Explore and Exploit labeled videos.**

that the TikTok algorithm attempts to personalize (exploit) a little over half the videos recommended to users. Accounting for the noise level of around 20% in `index-randomized`, we conclude that TikTok's algorithm exploits real users' interests in between 30% and 50% of all recommended videos in the first thousand videos of the users' tenure on TikTok.

*4.5.2 Distribution of Personalization Scores.* We compute the personalization score for the labeled videos in `real-tiktok`, and plot

the distribution of the scores in Figure 2. We observe that most exploit videos have a high personalization score (with a mean of 0.83), indicating that these videos are indeed specifically targeted and personalized to the users they are recommended to. Thus, when our framework identifies a item-user pair $(r_i, u)$ as an *exploit* recommendation, we can be confident that the exploit recommendation is actually the result of personalization in relation to the other users' timelines. On the other hand, we observe that most videos that are labeled explore have a much lower personalization score (with a mean of 0.08). This indicates that these videos are likely not personalized to the users they are recommended to since they are also often labeled as explore recommendations for other users.

We also investigate the distribution of personalization scores for exploit/explore videos with respect to the viewing duration, video popularity, and hashtag counts. We observed no correlation between personalization scores and video viewing and video popularity. However, we observe that exploit (explore) videos with fewer hashtags tend to have a higher (lower) personalization score (see Figure 3). This makes intuitive sense, since videos with more hashtags are often more generic [10], and hence harder to personalize, whereas videos with fewer hashtags often have more specific content that could be readily personalized to certain users.

## 5 RQ2: TIKTOK PERSONALIZATION FACTORS

In this section, we investigate the effect of different personalization factors on TikTok. We selected these personalization factors based on prior related work by Boeker and Urman [2], and the social media feed attributes available in our TikTok dataset. We examine two distinct user groups: the top quartile (TQ) and bottom quartile (BQ) of the mean user exploit fraction $\overline{\alpha}_i$, as computed in Section 4 on the `real-tiktok` dataset. We then examine the distributions of the factors to characterize the extent of personalization observed in each user group. Since $\overline{\alpha}_i$ is influenced by some of these factors, intuitively we expect the distributions to vary between the two groups. By comparing the difference in distributions of the factors between the groups, we elicit the importance of each factor on user personalization.

### 5.1 Experimental Setup

We first follow the experimental setup described in Section 4.4 to instantiate the model on the `real-tiktok` dataset. Then, we run our framework to label videos as exploit or explore. For each user, we calculate the user's exploit fraction, which is simply the number of exploit videos divided by the total videos recommended to that user, i.e., $N = 1000$. Next, we create two groups of users: 1) **Top Quartile, TQ:** users that are in the top quartile of the mean user exploit fraction (40 users with a mean user exploit fraction of 0.74); and 2) **Bottom Quartile, BQ:** users that are in the bottom quartile of the mean user exploit fraction (40 users with a mean user exploit fraction of 0.31).

For each user group, for each factor, we compare the distribution of that factor between the two user groups. Inspired by Boeker and Urman [2], we consider: 1) **Video watch percentage:** The mean video viewing duration percentage across all videos recommended to a user. 2) **Early skip rate:** The fraction of videos in the user's timeline skipped over very early, i.e., within 1 second. 3) **Fraction**

**Table 3: Factors influencing user personalization on TikTok. We report mean values for the BQ and TQ groups, $p$-values for the $t$ tests, and the impact level of each factor.**

| Personalization Factor | BQ | TQ | $p$-value | Impact |
|---|---|---|---|---|
| Watch Percentage | 84% | 93% | 0.03 | Medium |
| Early Skip Rate | 0.09 | 0.11 | 0.14 | Low |
| Fraction Liked | 0.03 | 0.14 | $10^{-5}$ | High |
| Fraction from Following | 0.02 | 0.3 | $10^{-14}$ | High |

**liked:** The fraction of liked videos in the user's timeline. 4) **Fraction from following:** The fraction of videos in the user's timeline that were uploaded by a creator that the user was following.

### 5.2 Results

We compare the distributions of each personalization factor across both the BQ and TQ user groups. Table 3 shows the different personalization factors studied in this section, and the level of impact each of these factors has in terms of driving the extent of content personalization for a user. We include violin plots to show the distributions of each factor in Figure 4, and also run the Student's $t$-test for statistical significance for each factor. We assign the level of impact to be "high," "medium," or "low" as characterized by the difference in distributions of the factor between the two user groups (a lower p-value corresponds to a higher impact level).

**Video watch percentage.** We observe a moderate difference between the BQ and TQ user groups in terms of their mean video watch percentage across all videos in these users' timelines. The TQ user group has a mean watch percentage of 93%, whereas the BQ group has a mean watch percentage of 84%. The $t$-test yielded a $p$-value of 0.03 indicating that there is a significant difference between the two groups (at the 0.95 confidence level). Additionally, we visually observe that the tail of the BQ distribution goes well below 50%, whereas the tail of the TQ distribution does not.

**Early skip rate.** We observe a marginal difference between BQ and TQ user groups in terms of the fraction of videos skipped over quickly, across all videos in users' timelines. The TQ has a mean early skip rate of 0.11, whereas BQ has a mean skip rate of 0.09. We observe outliers in both distributions, and $t$-test $p$-value of 0.14.

**Fraction liked.** We observe a significant difference between BQ and TQ user groups in terms of the fraction of videos liked. The TQ group liked 14% of videos on average, whereas the BQ group liked 3% of videos on average. Comparing the distributions from the violin plots in Figure 4c, we observe that the TQ group has a significantly higher fraction of liked videos above 0.2, whereas the BQ group (aside from a couple of outliers) lies entirely below this threshold. The $t$-test yielded a $p$-value of $10^{-5}$ indicating a very significant difference in distributions

**Fraction from following.** We observe a significant difference between the BQ and TQ user groups in terms of the videos in a user's timeline that were uploaded by a creator that the user was following. Visually, the distributions in Figure 4d differ greatly. We observed a $p$-value of $10^{-14}$ from the $t$-test, indicating a significant difference in distributions between the two groups.

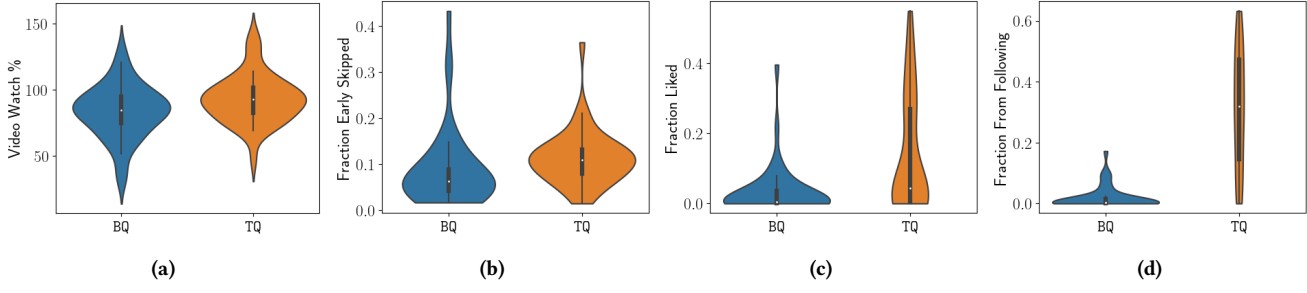

**Figure 4: Violin plots showing the distributions of personalization factors between the TQ and BQ user groups.**

# 6 CONCLUDING DISCUSSION

In this work, we proposed a framework that models personalization on social media feeds. We applied our framework to real data obtained from TikTok users and data obtained using automated accounts on TikTok, demonstrating our framework's applicability and validity. Our analysis shows that on TikTok, when considering the first one thousand videos in each user's timeline, the recommendation algorithm exploits the user interests in 30%-50% of the recommendation videos; this finding indicates that the TikTok algorithm opts to recommend a large number of explore videos in an attempt to either infer better the user interests or maximize user retention by recommending many videos that are outside of the user's (known) interests. Also, our analysis of the personalization factors finds that the most important aspects that affect the degree of personalization are following other TikTok accounts and liking videos. Our results (based on real TikTok user data) are in-line and confirm the results from Boeker and Urman [2], which made similar observations based on automated accounts. Our framework can assist various interested stakeholders in further understanding personalization on the Web. We elaborate on these use cases and the implications of our work below.

**For Platforms: Aiding Transparency Efforts.** Policymakers are currently demanding online platforms to provide end-users with explanations on why they are getting recommended specific content. We argue that our framework can be used to generate fine-grained explanations that can be used to inform the users why they are getting recommended specific content. For instance, assuming that the user liked a lot of videos with the #sports, a possible explanation could be "In the past 50 videos, you liked 20 videos with the #sports, so we inferred you liked sports content." We believe that our framework is a substantial step toward providing tools and techniques that can be used by online platforms to generate informative and precise explanations to end-users, hence being compliant with emerging regulations like the DSA [5].

**For Users: Insights into User Algorithmic Personalization.** Our framework can act as the backbone for future systems that end-users can leverage to extract insights into how personalized their social media feeds are. We envision that it is possible to implement an easy-to-use system where end-users can request their data from online platforms using the right of access by the data subject as described in the EU's General Data Protection Regulation [4]. Then, they can input their data into this system, which will leverage our framework to assess user personalization and then visualize to the

end users the extent of their personalization, which recommendation items are the result of personalization and which are not, as well as extract insights into what the online platform knows about them, through the lens of the content recommendations.

**For Policymakers and Researchers: Auditing Platforms and Algorithms.** We argue that our framework can assist policymakers and researchers in auditing online platforms and the effects of AI-based recommendation algorithms. Our framework is an important leap towards understanding the extent of personalization on other emerging platforms like YouTube shorts and Facebook/Instagram Reels. Such audits are of paramount importance, and policymakers can use the audit results to assess how compliant online platforms are with emerging regulations and act accordingly.

**Limitations & Future Work.** First, the sample of real traces from TikTok users is not necessarily representative; this is an inherent challenge that exists when undertaking studies that aim to recruit a small number of users from huge online platforms like TikTok. Nevertheless, we argue that the proposed framework and insights have merit and demonstrate that the degree of personalization varies across TikTok users. The fact that our findings match those of prior work that studied bot-based traces on TikTok [2] also increases our confidence. We note that the set of features we use is not an exhaustive list of all possible features but an intuitive set we curate to model the "relatedness" of items in most social media feeds. Consequently, we acknowledge that in some cases, our framework would only serve as a lower bound for the extent of personalization. Another limitation is that our analysis and results are agnostic to changes in the recommendation algorithm done by the platform over time. In this work, we audited the personalization of users regardless of when they started using TikTok. In the future, we plan to perform longitudinal audits to understand how the algorithm changes over time. Finally, our work lacks ground truth data on recommendation content that results from user personalization, another inherent challenge when performing such audits. To overcome this challenge, we use the notion of randomization and assume that there should be little personalization in randomized traces, hence evaluating our framework in this way.

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

# A PREPROCESSING TIKTOK RECOMMENDATIONS

We performed data preprocessing steps on the real users' TikTok video traces to model the sequence of recommended item-user pairs. Specifically, we filter out generic hashtags, we cluster hashtags into "topics" using word embeddings and clustering techniques, as well as extract the most popular hashtags for each user that act as the global interests of the user. We elaborate on our preprocessing steps below.

**(1) Filter generic hashtags.** On TikTok, there are certain "generic" hashtags that creators add to almost all videos in an attempt to influence the recommendation algorithm. Such generic hashtags include #foryoupage, #fyp, and #viral. These hashtags do not provide meaningful information and will likely affect our labeling of recommended videos as exploit or explore. Therefore, it is paramount to remove these common hashtags and ensure that our framework considers only meaningful hashtags. We follow a similar method to [2], inspect individual timelines, and filter out the most common hashtags for each user. Removing these generic hashtags helps ensure that videos are only related via meaningful hashtags specific to the videos' content.

**(2) Word2Vec hashtag clustering.** We use a heuristic fuzzy clustering method and Word2Vec [11] similarity to cluster sets of hashtags into similar "topics" to combine hashtags into comprehensive and concise groups and enable better matching of hashtags in practice. To do this, we first train a Word2Vec model, using Continuous Bag of Words, on all the video descriptions referenced in the entire dataset of real user traces (see Section 3.1). For training the Word2Vec model, we exclude words/hashtags that appear less than 10 times in the entire dataset and use a context window of 7.[1] Then, we use the cosine similarities of the Word2Vec embeddings of all the hashtags in our dataset. We then iteratively assign hashtags to the nearest cluster or create a new cluster based on the similarity of that hashtag with pre-existing clusters. For example the hashtags {#bieber, #biebertiktok, #belieber, #bieberforever} were all clustered into the same hashtag group. We use cluster centers to represent all hashtags in a particular cluster.

**(3) Extract popular hashtags.** For each user, $u$ we perform a term frequency-inverse document frequency (TF-IDF) analysis on the set of all hashtags corresponding to all videos recommended to that user (after removing the generic hashtags). Then, we compute the top-k hashtags (topics) of interest, as determined by the most relevant hashtags across all recommendations made to that user.

---

[1] We increase the context window to 7 instead of the default 5, because on TikTok several videos include a long list of hashtags.

