# OpenReview forum: "TikTok and the Art of Personalization: Investigating Exploration and Exploitation on Social Media Feeds"
_ACM.org/TheWebConf/2024/Conference — TheWebConf24_

### Official Review · Reviewer_Bqkm · 2023-11-18

**Novelty:** 5
**Technical Quality:** 2

**Review:**

The manuscript "TikTok and the Art of Personalization: Investigating Exploration and Exploitation on Social Media Feeds" presents an innovative approach to understanding personalization in social media feeds, with a focus on TikTok. The study introduces a novel framework for analyzing user timelines and applies this to real data, demonstrating the ability to audit and understand the recommendation algorithms used by TikTok.

However, the study has certain limitations that could be addressed in future work. Firstly, it primarily focuses on TikTok, and extending the analysis to other platforms like Instagram and YouTube Shorts could provide a more comprehensive view and enable valuable cross-platform comparisons. Secondly, the complexity of recommendation systems, often involving intricate neural networks and human-designed rules, suggests that observational analysis might not fully reveal their internal mechanisms. A more direct approach, possibly involving legal mandates for model transparency, could offer deeper insights.

Overall, the paper contributes significantly to the field of social media recommendation systems, offering a framework for auditing and understanding personalization. Enhancing the scope of the study and addressing its inherent complexities could further solidify its impact and relevance.

**Questions:**

1. The paper introduces an interesting research topic. However, from the perspective of a professional working in recommendation systems, it's important to note that these systems are composed of complex neural networks and various modules, and they also incorporate numerous human-designed rules. Therefore, it is quite challenging to understand the internal mechanisms of these systems solely through observational statistical data. A more direct approach might be to open-source the models in accordance with legal regulations for a thorough audit.

2. The analysis in the paper is limited to TikTok. It would be more convincing if the study, as mentioned in the introduction, also included an analysis of Instagram and YouTube Shorts, and provided a horizontal comparison across these platforms.

3. The section on Limitations & Future Work in the paper addresses very important issues. The authors are encouraged to further develop and elaborate on these points for a more comprehensive understanding.

**Reviewer Confidence:**

3: The reviewer is confident but not certain that the evaluation is correct

**Scope:**

4: The work is relevant to the Web and to the track, and is of broad interest to the community

---

### Official Review · Reviewer_VhNK · 2023-11-19

**Novelty:** 4
**Technical Quality:** 4

**Review:**

The paper presents a framework to analyse and measure the degree of recommendation of social networks and apply it to a dataset of Tik Tok users. To measure the degree of personalisation they propose a set of metrics to compare items engaged by users in a window of time to the items recommended in their timeline. The authors propose the framework to be used as an auditing tool which will become more important after the recent European Digital Services Act legislation. The authors also mention this framework could be used for explainability although it’s not the focus of the paper.

PROS:
* The topic of algorithm auditing is very relevant for practitioners in the area of personalization and recommender systems, specially after the upcoming DSA regulation.
* The presented framework is applied to a TikTok dataset with users that have given consent on donating their data.
* The paper is well written and clear.
* The proposed framework is simple but technically sound and can help in assessing the degree of exploitation vs exploration of recommendations in social networks.
* Findings match those of related work that uses automated accounts instead of real accounts.

CONS:
* The paper is focused on a specific social network which is TikTok and uses a small dataset of users (347 users filtered down to 220). This is a reasonable limitation when dealing with donated data but also makes findings less strong.
* Related work is focused on analysing and understanding the TikTok algorithm but misses on citing work related to algorithm auditing frameworks which is the main contribution of the paper.
* The framework only measures the degree of personalization that users receive in social networks but misses important aspects such as gender or race biases which are important in ethical algorithmic auditing.
* No code is submitted which makes the framework harder to apply.

Overall the topic of algorithm auditing is very relevant and work such as this is needed. This paper presents an auditing framework that aims to detect the degree of personalisation of algorithms (i.e. exploitation vs explorations). However, personalisation in itself is not a problem that concerns regulators, but more its possible adverse effects, such as promoting negative or biased content that could lead to people’s mental health issues like depression or reinforcing discrimination towards other people and beliefs. Although this framework would be a starting point it does not directly address these issues. In this sense, this work covers just one side of algorithmic auditing but misses important parts.

**Questions:**

Issues that would make the paper stronger:
* Include related work about algorithm auditing frameworks and compare their solution to theirs.
* Submit code of their framework.
* Extend the framework by incorporating other important aspects for algorithmic auditing such as gender or race bias or detection of negative topic reinforcement.

**Ethics Review Description:**

No issues.

**Reviewer Confidence:**

3: The reviewer is confident but not certain that the evaluation is correct

**Scope:**

3: The work is somewhat relevant to the Web and to the track, and is of narrow interest to a sub-community

---

### Official Review · Reviewer_cQ32 · 2023-11-22

**Novelty:** 3
**Technical Quality:** 5

**Review:**

Pros
1.	The researcher delves into the degree of personalization on TikTok, which is currently the most popular short video recommendation platform. This topic is both intriguing and relatively underexplored.
2.	With real-world TikTok dataset to, the authors propose a framework for measuring the level of personalization on short video recommendation platforms. This framework can potentially be applied to other platforms as well. The conclusions drawn from the study hold significance in the field.

Cons
1.	The author's experiment suffers from a lack of user samples, potentially leading to biased results.
2.	The features selected by the authors may not align with the features used by the actual TikTok recommendation system. As a result, it becomes challenging to directly determine whether a video is considered "explore" or "exploit."

**Questions:**

1.	Most recommendation systems rely on neural networks, which lack interpretability. How can we assess whether a video falls under the "explore" or "exploit" category?
2.	In comparison to BQ, TQ shows a slight increase in the early skip rate. This finding contradicts intuition, as one would expect users to skip videos more frequently when personalization is high. Could you provide a reasonable explanation for this?

**Reviewer Confidence:**

3: The reviewer is confident but not certain that the evaluation is correct

**Scope:**

4: The work is relevant to the Web and to the track, and is of broad interest to the community

---

### Official Review · Reviewer_cLes · 2023-11-26

**Novelty:** 5
**Technical Quality:** 5

**Review:**

The problem addressed is very interesting: figurin out which recommendations are personalized and which not. However,

RQ2 is not well phrased. You cannot use "certain factors", it's too vague. Be specific!

The technical part is not easy to follow. From what I understood, the authors created the ground truth by understanding if the recommended item is related to the previous user behaviour. However this relation has been arbitrarily seected by the authors. Hence a different "activation" condition would yield different labels.

I still can't understand how the ground truth labels have been collected, if at all.

Other than the missing info about data labeling I think the work has important implications as the users elaborated in the discussion.

Update: the authors have answered but could not provide improvement over the issues I raised. I am keeping my score.

**Questions:**

NA

**Reviewer Confidence:**

3: The reviewer is confident but not certain that the evaluation is correct

**Scope:**

4: The work is relevant to the Web and to the track, and is of broad interest to the community

---

### Official Review · Reviewer_Wkc9 · 2023-11-29

**Novelty:** 7
**Technical Quality:** 5

**Review:**

The paper explores personalization in social media feeds, with a focus on the TikTok platform. Concretely, the study presents a framework that helps distinguish between exploration and exploitation in users' timelines whereas exploitation is defined as the algorithm creating recommendations based on the user's preferences and previous actions and an exploration recommendation is a recommendation that is not the result of user personalization but derived from the algorithm exploring if a user might like an item.

Pros
- The evaluation setup is interesting and based on user trails and bot-generated trails.
- Ethics are clearly described and addressed.
- Clear and comprehensive methodology
- Findings are interesting and make a significant contribution to the field (in particular, it is interesting to see that the algorithm of TikTok seems to exploit user interests in up to 50% of the recommended videos).
- The introduced personalization score is a good tool to quantify the extent of personalization.
- Findings are embedded in a broader discussion about transparency and privacy, and implications of the study for various stakeholders are addressed.
- The paper explicitly mentions limitations, including that the sample used for the study is not representative.

Suggestions for improvement:
- The paper leverages a non-representative sample of TikTok users (which is acknowledged in the paper), which raises questions related to generalizability of the findings.
- The distinction between exploration and exploitation is not well backed up with theory and lacks intuition. More clarity is advisable here.
- Integrating qualitative aspects through user interviews or surveys could help understand user perceptions and experience of the impact of personalization and "exploration recommendations".
- It is not well described how recommendations are labeled into exploration and exploitation - also the used local and global features are introduced only briefly and not well motivated. They seem to be very specific for the TikTok platform, and the question is what features would be needed in other similar platforms.
- The study is conducted only on TikTok, and it is not clear if findings translate also to other platforms.
- Some design choices and parameter settings lack a motivation (filtering out users, window size, amount of hashtags,..).
- A deeper discussion of ethical implications on having up to 50% of recommendations coming not directly from user behavior, would be valuable (e.g., related to impacts on user behavior, etc.)
- Figures are in too low quality and need to be revised
- No publicly available implementation is described, which limits the reproducibility of the work.

Assessment after the rebuttal:
- The authors clarified several issues and I updated my scores based on these clarifications.

**Questions:**

- How did you come to the definitions of exploration and exploitation?
- Can you provide more details on how the recommendations were labeled into exploration and exploitation?
- Do the findings translate also to other, similar platforms?
- What ethical implications do you see with having up to 50% of recommendations not directly resulting from user behavior?
- Can you provide justifications for parameter settings and design choices (see above)?

**Ethics Review Description:**

Relevant ethics are discussed in the paper.

**Reviewer Confidence:**

3: The reviewer is confident but not certain that the evaluation is correct

**Scope:**

4: The work is relevant to the Web and to the track, and is of broad interest to the community

---

### Decision · Program_Chairs · 2024-01-22

**Decision:**

Accept

**Comment:**

The paper studies explore-exploit trade offs when personalizing the TikTok feed.

 Reviewers appreciated many aspects in this paper. It explores an important topic that touches upon everyday life routines of many people.
 Methodology is sound at most, many reviewers commented on the code, and the authors have responded. I appreciate some data and parameters cannot be disclosed and it seems the authors do a good job providing transparency in other means, given this limitation.
 many reviewers agreed the paper presents intriguing findings, which are of interest to the field, as well as interesting concepts, such as the personalization score.
 the limitation section also received positive feedback.
 While the paper has weaknesses, such as lack of information about sensitive parameters, design choices, and the fact it is conducted only on TikTok, I think it offers enough merits to warrant publication.
 Authors are asked to make sure their code is public. And please address the minor-change comments from reviewers if accepted.